# The Transmission of SARS-CoV-2 Infection on the Ocular Surface and Prevention Strategies

**DOI:** 10.3390/cells10040796

**Published:** 2021-04-02

**Authors:** Koji Kitazawa, Stefanie Deinhardt-Emmer, Takenori Inomata, Sharvari Deshpande, Chie Sotozono

**Affiliations:** 1Department of Ophthalmology, Kyoto Prefectural University of Medicine, Kyoto 6020841, Japan; csotozon@koto.kpu-m.ac.jp; 2Buck Institute for Research on Aging, Novato, CA 94945, USA; SDeshpande@buckinstitute.org; 3Institute of Medical Microbiology, Jena University Hospital, 07743 Jena, Germany; 4Department of Ophthalmology, Juntendo University Graduate School of Medicine, Tokyo 1138421, Japan; tinoma@juntendo.ac.jp; 5Department of Strategic Operating Room Management and Improvement, Juntendo University Graduate School of Medicine, Tokyo 1138421, Japan; 6Department of Hospital Administration, Juntendo University Graduate School of Medicine, Tokyo 1138421, Japan; 7Department of Digital Medicine, Juntendo University Graduate School of Medicine, Tokyo 1138421, Japan

**Keywords:** SARS-CoV-2, COVID conjunctivitis, COVID-19, ACE2, virus entry, ocular transmission

## Abstract

The severe acute respiratory syndrome coronavirus 2 (SARS-CoV-2) is a global health problem. Although the respiratory system is the main impaired organ, conjunctivitis is one of its common findings. However, it is not yet understood if SARS-CoV-2 can infect the eye and if the ocular surface can be a potential route of SARS-CoV-2 transmissions. Our review focuses on the viral entry mechanisms to give a better understanding of the interaction between SARS-CoV-2 and the eye. We highlighted findings that give evidence for multiple potential receptors of SARS-CoV-2 on the ocular surface. Additionally, we focused on data concerning the detection of viral RNA and its spike protein in the various ocular tissues from patients. However, the expression level seemed to be relatively low compared to the respiratory tissues as a result of a unique environment surrounding the ocular surface and the innate immune response of SARS-CoV-2. Nevertheless, our review suggests the ocular surface as a potential route for SARS-CoV-2 transmission, and as a result of this study we strongly recommend the protection of the eyes for ophthalmologists and patients at risk.

## 1. Introduction

The ongoing pandemic caused by the severe acute respiratory syndrome coronavirus 2 (SARS-CoV-2) is a global health problem that the World Health Organization (WHO) declared as the coronavirus disease 2019 (COVID-19) pandemic on 11 March 2020 [1]. Since the outbreak of the pandemic, 2.6 million people have died worldwide. The ongoing situation is reinforced by emerging viral mutations, leading to increased infectivity of SARS-CoV-2. In addition, superspreading events also drive the pandemic with a continued discussion of transmission processes.

The main clinical symptoms of COVID-19 are mild to severe respiratory illness including fever, dry cough, running nose, and shortness of breath. However, conjunctivitis is one of the common symptoms in COVID-19 patients [1,2,3]. Conjunctivitis is a common disease that affects tens of millions of people worldwide each year. In the United States of America, it is estimated that 1% of primary care physician visits are related to conjunctivitis [4]. Adenoviral conjunctivitis is the most frequent manifestation of viral conjunctivitis. SARS-CoV-2 is also considered to be another differential diagnosis of viral conjunctivitis.

In the middle of the COVID-19 pandemic, various healthcare systems in ophthalmology including routine patient care and surgeries were greatly affected. Although various ophthalmological societies have issued their statements, there are still no universally agreed upon guidelines [5]. For instance, it is essential for ophthalmologists to know whether donor corneas obtained from COVID-19 patients or asymptomatic patients can be available for corneal transplantation. Therefore, an understanding of the viral replication and transmission is essential for a proper diagnosis and standard precaution.

Our review highlights the viral entry mechanism of the ocular surface as an essential part of the pathogenesis of conjunctivitis. For this, we described the infectious pathway of SARS-CoV-2, which belongs to the family of RNA viruses, and we investigated reports of transmission and disease course in the eye. Although many reports of SARS-CoV-2 have been published, there are still limited data with reports on the eye, and it is controversially debated if SARS-CoV-2 can infect the ocular surface and transmit via the ocular fluid [6]. Napoli et al. suggest that the virus is capable of replicating in the conjunctiva [6]. Therefore, the wearing of eye protection is strongly recommended to prevent transmission. In our review, we discuss the most recent studies on this topic.

Furthermore, we illustrate the viral structure and highlight the known receptor and coreceptors of SARS-CoV-2 to explain the mechanism of the ocular tropism, and transmission routes. Finally, we also present the available literature with regard to feasible therapeutic strategies.

Our review shows that the existence of the molecular structures on the ocular surface has been demonstrated. Even when the replication at the ocular surface is described as low, a possible transmission could be expected and a precaution against possible transmissions is strongly recommended.

## 2. Ocular Signs of COVID-19 Patients

Viral conjunctivitis causes hyperemia of the bulbar and palpebral conjunctiva due to an inflammation in the conjunctiva, and increases eye discharges, tearing, itchiness, irritation, and gritty feelings on the ocular surface [7]. Severe symptoms lead to conjunctival edema, swelling of the eyelid, or as part of general cold symptoms, with swollen lymph nodes anterior to the ear, fever, a sore throat, and runny nose.

The incubation period of COVID-19 is about 1 to 14 days, and the WHO reports that the average time from infection to the onset of symptoms is about 5 to 6 days. Ocular symptoms usually appear after the onset of respiratory symptoms. However, it should be noted that prodromal ocular symptoms occurred in approximately 10% of COVID-19 patients [8,9,10], and delayed onset of conjunctivitis after severe COVID-19 infection was observed in some cases [11,12,13].

Ocular signs in patients with SARS-CoV-2 infection are still not well-described. It has been almost a year since the WHO declared the COVID-19 pandemic, and the evidence of ocular manifestation in COVID-19 patients is gradually accumulating. Our current systematic review and meta-analysis, with 15 studies involving 1533 patients with SARS-CoV-2, revealed that 11.2% of patients had ocular symptoms, mostly presenting with conjunctivitis (86.4%), followed by ocular pain, dry eye, and floaters [8]. Large retrospective clinical studies regarding SARS-CoV-2 do not include detailed ophthalmic examinations, because patients with SARS-CoV-2 present with life-threatening clinical scenarios [1]. In general, SARS-CoV-2 does not appear to attract significant reactions from the immune system at the ocular surface and is thus rarely associated with inflammatory responses of the ocular surface, unlike with the respiratory system [14]. The natural history of COVID-19 seems to be rapid self-limited conjunctivitis that improves without any treatment and does not affect visual acuity, nor is associated with short-term complications [15]. However, Cheema et al. reported that one patient initially presented with herpes-like pseudodendritic infiltration on the cornea, leading to a decline in vision due to severe keratoconjunctivitis [16], and relapsing viral keratoconjunctivitis and conjunctivitis with pseudomembranous also have been reported [12,17]. The corneal sensitivity testing assessed by the Cochet–Bonnet aesthesiometer did not seem to change in COVID-19 patients, unlike with herpes viruses [18]. Case reports with episcleritis [19,20] have been reported as well, suggesting that SARS-CoV-2 can be involved in the cornea and sclera as well as in the conjunctiva.

Recent reports revealed that the retina might also be affected post SARS-CoV-2 infection [21,22]. Detailed analyses with optical coherence tomography (OCT) demonstrated hyperreflective lesions at the level of ganglion cell and inner plexiform layers more prominently at the papillomacular bundle [22]. However, OCT angiography was reportedly normal, suggesting that it can raise some questions as to whether these findings are pathological conditions. Another case with retinal vein occlusion (RVO) in a COVID-19 patient has been recently reported [21]. This case is a 52-year-old male and does not have any underlying medical condition to possibly cause RVO, such as diabetes, hypertension, or tuberculosis, indicating RVO occurs secondary to SARS-CoV-2 infection. Recent more papers demonstrated several cases with central RVO [23,24,25,26]. However, it is still unclear whether it is caused by the direct SARS-CoV-2 infection to the retina or by thrombosis in the retinal vascular vessels due to the cytokine storm associated with COVID-19. Further, the intense inflammation may drive the severe allograft rejection after corneal transplantation, which required repeat penetrating keratoplasty (PK) [27]. Chorioretinitis from fungal sepsis has also been reported [18]. The severe COVID-19 patients hospitalized in an intensive care unit (ICU) setting may be attributed to secondary fungus endophthalmitis [18]. Case reports with acute macular neuroretinopathy [28,29], optic nerve infarction [30], and cerebral venous thrombosis [31,32] have been reported as well. Thus, the inflammatory-associated responses secondary to COVID-19 should be paid careful attention.

## 3. Viral Structure of SARS-CoV-2

Viruses are characterized by a simple structure composed of a viral genome and a nucleocapsid. The capsid serves as a protein coat to protect the viral genome, organized in a helical or spherical manner. It is assembled by single protein components, defined as capsomers. However, some viruses are composed of both shapes and are therefore referred to as complex viruses.

The nucleic acids contain either single- or double-stranded RNA or DNA, forming together with the protein-composed capsid the nucleocapsid [33]. An optional structure is a viral envelope, assembled by a phospholipid bilayer with glycoproteins. This structure originates from the nuclear or plasma membrane of the host cell. Within this common microstructure, the particles differ remarkably in size, form, and shape [34]. In our review, we describe *Coronaviridae* in the following section, particularly focusing on the structures involved in the entry mechanism.

Members of the family *Coronaviridae* are enveloped, single-stranded RNA-viruses. Despite the vulnerability of enveloped viruses concerning environmental resistance, human coronaviruses (HCoV) are relatively highly resistant to destruction due to temperature and humidity [35]. This characteristic is a major contributor to the current COVID-19 pandemic, the outbreak of SARS-CoV-2, which is expected to have occurred in Wuhan, China in 2019 [36]. The family of *Coronaviridae* is further classified into the subfamilies *Coronavirinae* and *Torovirinae,* revealing the largest known RNA genome (25–32 kb). Characterized by a spherical virion structure, the virus envelope is composed of glycoproteins. Here, the spike (S) protein is the major protein responsible for the typical appearance of the virus. In total, Coronaviruses encode for 4 proteins. In addition to the S protein, further proteins are involved in the formation of the membrane, named membrane (M) and envelope (E) proteins. The nucleocapsid (N) protein forms the flexible nucleocapsid [37].

In general, infections by HCoV lead to respiratory diseases, with symptoms of the common cold and upper respiratory tract infections. However, since the outbreak of the severe acute respiratory syndrome (SARS) virus in 2002, the fatal potential became evident. However, even case reports of keratoconjunctivitis have been reported, which are likely to be more common and could be an important early detection symptom [17]. The emerging variant SARS-CoV-2 presents a significantly increased infectivity compared to SARS-CoV. One of the major explanations for this is a mutation of the S protein [38]. Recently detected novel variants of SARS-CoV-2 show mutations at the spike protein leading to higher infectivity of the virus [39]. Independent authors have determined that there is an increased viral load triggered by the D614G mutation [40,41]. Currently, several mutations are identified, and the risk of a further exacerbation of the pandemic due to the increased infectivity is possible.

## 4. Method

An extensive search strategy was designed to retrieve all articles published from March 1, 2020 to March 1, 2021, by combining the genetic terms, ((coronavirus 2019 OR COVID-19 OR COVID OR SARS-CoV-2 OR (sars cov 2) OR (2019 novel coronavirus) OR 2019-nCoV) AND (Conjunctivitis OR (Ocular Surface) OR Eye OR Ophthalmology) AND (viral entry OR receptor)) in key electronic bibliographic databases (PubMed and EMBASE).

Exclusion criteria were as follows: (1) Clinical guidelines, consensus documents, reviews, systematic reviews, and conference proceedings, (2) studies about other serotypes of severe acute respiratory syndrome coronavirus and Middle East respiratory syndrome coronavirus infection, (3) mental health status during COVID-19, (4) preprinted articles, (5) conference abstracts. Three independent investigators (K.K., S.D.-E., S.D.) conducted a literature search and removed the duplicates. The full texts of all papers were assessed individually. Eventually, 56 articles were selected for our review.

## 5. The Ocular Tropism of SARS-CoV-2

The entry process of the viral infection of SARS-CoV-2 is a complex mechanism regulated by various co-receptors with high cleavage activity. During the entry process, the receptor-binding-protein (RBD) of SARS-CoV-2, which is located between the S1 and S2 subunit of the spike protein (S), binds with a high affinity to angiotensin-converting-enzyme 2 (ACE2). Afterwards, the spike protein is cleaved into the S1 and S2 subunits [42], which allows the S2 unit a closer contact to the host membrane (S2 = fusion protein). The peptidase activity of ACE2 is necessary for interaction with the host membrane [43]. However, many different co-receptors with cleavage activity are described in the literature as being responsible for this necessary step. Initially, furin leads to a conformational change of the spike protein followed by TMPRSS2, described as particularly relevant for this process. In addition, ACE2 triggers the cleavage of ADAM17, which is an essential proteinase.

In general, a protein on the surface of the virus targets one of the receptors exposed on the surface of the host cell for attachment. Afterward, the intracellular entry is mediated by the (1) plasma membrane, the (2) early, (3) maturing, and (4) late endosome, the (5) macropinosome, and/or (6) the endoplasmic reticulum [44].

The ocular tropism of SARS-CoV-2 is still unclear [14]. Single case reports of HCoV-associated conjunctivitis were published [17] and some authors describe ocular symptoms, especially in critically ill patients [3]. Interestingly, patients with conjunctivitis show detectable viral RNA in tears by quantitative reverse transcription polymerase chain reaction (qRT-PCR) [45,46,47], while patients without signs of conjunctivitis did not show viral RNA in ocular specimens. This suggests that the eye is not the classical route of SARS-CoV-2 infection. Whether SARS-CoV-2 is detectable at an early stage via the tear fluid is still unclear, and more studies are necessary to systematically analyze the viral load.

To understand the ocular tropism of SARS-CoV-2, we present the molecular structures of the cornea, the conjunctiva, and the retina, which could be involved in the process of the viral entry.

The ACE2 is confirmed as the primary receptor of SARS-CoV-2 [48] and predominantly present in corneal and conjunctival cells [49,50,51]. Interestingly, retina is a tissue which highly expresses ACE2. These results were proven by analyzing ACE2 expression across different tissues on the mRNA level [52,53]. In addition, the receptor was also determined directly via immunofluorescence staining, both in the cytoplasm as well as on the cell membrane [54]. The main function of ACE2 is the control of the blood pressure via the regulation of blood volume [55]. However, components of the renin-angiotensin system could also be detected in various components of human eyes, yet the function remains unclear [56]. The virus entry is enhanced by overexpression of ACE2, and Monteil et al. recently showed the efficient inhibition of a SARS-CoV-2 infection by using recombinant ACE2. Here recombinant ACE2 blocks SARS-CoV-2 in vitro [57,58,59]. However, studies show that downregulation of ACE2 leads to a proinflammatory process in the lung [59]. It is hypothesized that ADAM17 is relevant for the fusion of SARS-CoV-2 with the host membrane. On the other hand, it is assumed that the shedding of ACE2 leads to a lower ACE2 membrane form and, from this, to a reduced virus entry [60].Furin belongs to the family of type I transmembrane serine-proteases. The enzymatic activity is connected to the subtilisin-like catalytic domain [61,62]. The role of furin as an activator of various substrates is well investigated [63]. Collin et al. prove the expression of furin at the conjunctiva and at the corneal and limbal epithelium [64], suggesting a potential entry of SARS-CoV-2 at the ocular surface.The transmembrane serine protease 2 (TMPRSS2) is responsible for priming of the S protein and therefore is very important for host entry. Hoffmann et al. published their findings on the blocking of a SARS-CoV-2 infection by using serine protease inhibitors, a method that emphasizes the relevance of this protease [65]. Zhou et al. were able to detect TMPRSS2 on various specimens of the eye [66], and TMPRSS2 is reported as highly expressed in the murine cornea, compared to the conjunctiva [67,68]. Recently published data show that ACE2 and TMPRSS2 is co-expressed at the human conjunctiva [69].The disintegrin and metalloproteinase domain 17 (ADAM17) is an important proteinase, expressed by various tissues [57]. It is responsible for the shedding of protein ectodomains and is popular for its shedding of TNFalpha. Additionally, ACE2 is also shed by ADAM17 and upregulated by angiotensin II via type 1 receptors (AT1). However, Palau et al. mentioned the controversial debate about ADAM17 and suggested a protective effect on COVID-19 of ADAM17 inhibition. Altogether, the impact of ADAM17 on SARS-CoV-2 needs to be clarified to understand how ADAM17 regulates the viral entry [70]. In general, ADMA17 is known to be relevant for the morphology of the eye, proven in mouse experiments [71].CD147 (HAb18G, basigin, EMMPRIN) is a member of the transmembrane glycoproteins of the immunoglobulin superfamily. Pushkarsky et al. described CD147 as a crucial cofactor for HIV infection, demonstrating a high replication capacity at early stages of infection. Recently published data suggest an interaction of CD147 with SARS-CoV-2 [72]. The authors were able to show that SARS-CoV-2 enters the cell via CD147-mediated endocytosis. This clathrin-independent endocytosis was also described in a mouse model with humanized CD147 mice [72]. Concerning human eyes, the receptor could already be detected in the cornea, the conjunctiva, and in ocular fluids [73,74].Cathepsin L (CTSL) is a cysteine protease recently described as involved in the establishment of COVID-19. This lysosomal enzyme is crucial for the initiation of protein degradation, e.g., for the L-lactate dehydrogenase (LDH) [75]. For SARS-CoV-2 a cleavage of the S1 unit has been reported [76]. The human cornea expresses CTSL and is reported to be relevant for corneal angiogenesis [77]. CTLS-inhibitors were mentioned as a potential treatment strategy in COVID-19 patients and could be used as a combination therapy as a useful strategy to block the host cell entry.The dipeptidyl peptidase-4 (DPP4) belongs to the family of type I transmembrane proteins, available in the soluble and transmembrane form. DPP4 has a high cleavage potential and therefore various substrates are known to be cleaved. In addition to the Glucagon-like-peptide (GLP-1), additional proinflammatory cytokines and chemokines were inactivated by DPP4 (e.g., RANTES, IP-10, HMGB1) [78]. DPP4 was described as a main receptor for the Middle East respiratory syndrome (MERS) in 2012, and during the early days of the ongoing pandemic it was described as a coreceptor able to bind to the spike protein of SARS-CoV-2 [79]. DPP4 is widely expressed and also described in the retina [80] and the ocular surface [64].

In conclusion, these molecular findings indicate that the viral entry of SASR-CoV-2 on the ocular surface is feasible. In Figure 1, we highlight molecular findings of the entry mechanism, based on experimental or clinical data in mouse and human models.

## 6. Modes of Transmission and Prevention of Viral Infections

In recent years, different transmission pathways of viruses have been proposed [81]. In general, the transmission of enveloped viruses (*Coronaviridae*) is attributed to direct and indirect contact, respiratory droplets, and airborne transmission [82,83]. Unlike non-enveloped viruses (i.e., *Adenoviridae*), enveloped viruses are easily inactivated by routine surface cleaning and disinfection. Appropriate personal protection should be taken for those responsible for the decontamination of a room or area. In the subsequent chapter, the transmission routes of coronaviruses are highlighted separately, followed by a discussion of prevention strategies.

### 6.1. Transmission Routes of SARS-CoV-2

The main transmission pathway of SARS-CoV-2 is the respiratory tract. However, extrapulmonary routes for transmission have also been described in the literature. In addition to respiratory droplets, contact transmissions by touching contaminated surfaces also seem to play a role, as data on the infection of mucous membranes have recently been published [84]. The spread of infection through the hands seems to play a significant role, particularly triggered by fecal infection. An observational study showed that people touched common surfaces and their mouth or nose mucosa at a rate of 3.3–3.6 times per hour [85]. This suggests that, even if the chances of infection are small, the ocular surface may become an entry point for the virus and a reservoir for viral infection. Our previous systematic report indicated that a viral infection is possible through the eye route [8]. Here, the testing of conjunctival swabs was positive in 16 out of 60 cases (27%) [8], and SARS-CoV-2 RNA was detected in tears and conjunctival secretions [45]. In addition, RNA in the tear fluid of patients with moderate to severe COVID-19 was collected with a conjunctival swab or the Schirmer test, and the differences depending on the collection method were examined. However, the collection rate was similar in either method, and the detection rate of viral RNA in the tears reached 24% [47], while no viral RNA was detected in the tear fluid in COVID-19 patients [86,87]. Furthermore, the investigation of human post-mortem ocular tissues in COVID-19 patients found that there was no SARS-CoV-2 RNA in the different parts of ocular tissues and intraocular fluid including corneal epithelium, stroma and endothelium, conjunctival epithelium and fluid, and the anterior chamber [88]. Although we cannot deny if the eye is the transmission route, it suggests that the eye does not seem to be a common transmission route.

However, the majority of COVID-19 studies did not report and investigate the use of eye drops, which could have a potent antiviral effect. In fact, the systematic review by Napoli et al. revealed that many of the eye drops routinely used in ophthalmology, even artificial tears, have antiviral effects, and it is necessary to discuss whether or not the use of anti-glaucoma and anti-allergy eye drops as well as antibiotics eye drops, should be investigated [89]. More recent reports also analyzed donor ocular tissues in COVID-19 patients, and SARS-CoV-2 RNA was detected in 17 out of 132 samples (13%), including cornea and sclera samples, and the SARS-CoV-2 enveloped protein was also detected in the cornea of the COVID-19 donors [90]. Another report also showed SARS-CoV-2 genomic RNA detected in the cornea [91]. Chen et al. described secondary conjunctivitis after 13 days of respiratory illness due to COVID-19 [13]. Here, the occurrence of bilateral conjunctivitis with watery eyes was impressive. The authors strongly suggest the importance of caution when performing an ophthalmic examination. Interestingly, the quantification of SARS-CoV-2 RNA expression was determined with cycle threshold (Ct) values, and RNA in a COVID-19 patient at acute phase was detected at 31 cycles; Ct values were decreasing as their ocular finding improved [13]. Furthermore, RNA obtained from the COVID-19 donors was detected at 29 to 35 cycles [90], which were relatively lower Ct values compared to those of nasal and throat [92], suggesting that viral loads in the ocular surface do not seem to be high. Although only high crossing points were detected in the qRT-PCR of the conjunctival swabs, transmission by this route would be possible.

In addition to the onset of ocular symptoms during a COVID-19 course of infection, primary ocular-triggered infection is also possible. Deng et al. have demonstrated that the conjunctival infection of rhesus macaques leads to pneumonia [93]. Interestingly, this infection route resulted in mild pneumonia with a lower viral load compared to an intratracheal infection. This indicates that the virus can indeed spread from an ocular infection in a generalized manner throughout the entire organism.

These findings correspond to the expression of ACE2 and TMPRSS2 on the conjunctival surface [52,66]. The results of Zhou et al. are in contrast to other studies, demonstrating that ACE2 and other entry factors of SARS-CoV-2 including ENPEP, ANPEP, DPP4, and TMPRSS2 are not substantially transcribed in conjunctival tissues [94]. However, since the infection has now been proven, it seems correct that the conjunctiva shows a lower expression rate of essential receptors than does the respiratory tissue. Nevertheless, the drainage of tears through the nasolacrimal gland could be particularly involved in this process [86].

### 6.2. Prevention of Viral Transmission Triggered by Ocular Infections

SARS-CoV2 is a highly contagious virus. Since viruses can be transmitted in various ways, a spread of infection through the eye should not be excluded. Social distancing is the most reliable precaution. Wearing masks by people, including all the time by patients, can help limit the spread of the virus that causes COVID-19. The Centers for Disease Control and Prevention (CDC) recently proposed that wearing a cloth mask on top of a surgical mask helps improve the fit of the surgical mask, a technique called double masking [95]. It is also useful to limit the number of patients in a room who might spread the virus, by an arrangement of their medical appointment schedules, and to screen patients who have any COVID-19 symptoms or potential SARS-CoV2 exposure history, including travel and contact with someone who has been diagnosed with COVID-19 within the previous 14 days. As provided in the guidelines of the American Academy of Ophthalmology and other ophthalmology societies, social distancing in waiting rooms, frequent disinfection, and mandatory mask use have been suggested [5]. However, there is still the lack of universally agreed recommendations on safety systems and legal protection for clinical ophthalmologists, and it is essential to accumulate the further evidence in the future.

The instruments for eye examinations, including slit lamps and their accessory lenses, have also been known to be a source of viral transmission [96]. Among eye examinations, non-contact tonometry may pose a risk for transmission of the SARS-CoV-2 due to air pressure to measure intraocular pressure (IOP), which is one of the most frequent tests in daily practice. Various academic societies have made recommendations on its use and measures to be taken [89]. For instance, the EuroTimes recommends avoiding the use of non-contact tonometers, and using contact tonometers with disposable tips. The American Academy of Ophthalmology allows contact tonometers with reusable tips with 70% ethanol disinfection, but basically recommends the use of disposables.

In the clinical setting, the ophthalmic examination room is limited, and ophthalmologists are quite close to the patients whenever they see the patients. An experimental simulation study revealed that simulated coughing spread droplets throughout the entire examination room. Respiratory droplets were reduced with masking, but were still observed on the slit lamp joystick and the slit lamp shields [97,98]. Based on the surface area of the slit lamp, it was found that about half of the area was contaminated by the patient’s exhaled air [99]. Since SARS-CoV-2 RNA in tears and conjunctival secretions of COVID-19 patients was detected [45,47], careful attention should be paid to the spread of viruses via ophthalmic instruments, towels, doorknobs, etc., because large amounts of the virus could be contained in eye discharges and tears during viral infection [100].

In the beginning of the pandemic, corneal transplant surgeries were greatly affected due to a large shortage of donor corneas [101]. In Europe, donor corneas decreased by 38%, 68% and 41%, respectively, during March–May 2020 compared with those in last 2 years [101]. In India, eight out of the 20 eye banks did not collect corneal tissue from April to June 2020, and the number of transplants dropped by two-thirds [102]. In addition, more than half of the surgeons decided not to perform oculoplastic surgeries in consideration of the COVID-19 pandemic, and about 1/4 of the surgeons performed only emergency surgery [103]. Delays in vitreous injections for the treatment of retinal diseases also occurred, with a delay of more than two months and the need for three or more injections in the past being significant poor prognostic factors for visual outcome in diabetic macular edema patients and macular degeneration [104,105]. It has been pointed out that cataract surgeries may also be discontinued or postponed during a pandemic and that this may lead to long-term retention of patients [106], and we need to consider measures to move healthcare appropriately even during a coronavirus pandemic.

There are several experimental reports on the transmission of SARS-CoV2 during cataract surgery. During phacoemulsification and irrigation/aspiration, droplets from the intraocular reflux fluid may adhere to the surgeon’s gloves and gown [107]. However, in a study of visualization of aerosols and droplets during cataract surgery in real-world settings, no visible aerosolization was detected, and droplets were detected but not by an water indicator, confirming direct contact [108]. This suggests that although droplets do occur, they do not pose a significant infection risk to the surgeon.

The importance of eyeglasses has also been proposed. A member of the national expert panel on pneumonia in Wuhan, China, who was infected by COVID-19, wore an N95 mask but did not wear anything to protect his eyes [109], and the Chinese ophthalmologist who was working on the treatment of COVID-19 patients later passed away. Another report revealed that patients wearing glasses constituted a lower percentage of patients who were hospitalized than did the general population [110], suggesting that a face shield and a proper shield between patients and ophthalmologists can protect people from and help to limit the SARS-CoV2 transmission. However, the wearing of eye protection may not be beneficial only for the treating physicians. Napoli et al. suggest that particularly immunocompromised patients can benefit enormously [6].

The prevention of the spread of SARS-CoV-2 from pre-symptomatic patients is essential, especially since tear fluid could be contagious before symptoms arise. In fact, in outpatients with no COVID-19 symptoms who received ophthalmic examinations in the same room as they usually do, the qRT-PCR testing for SARS-CoV-2 on the tables and ophthalmic instruments revealed two positive samples, which were taken from the slit lamp breath shield and the phoropter, respectively [111]. Since it is challenging to remove viruses from the skin and environmental surfaces, frequent hand hygiene is important for patients and for those who may have come into contact with items to which the virus is attached. Regular handwashing and surface disinfection of instruments for an eye examination with 0.1% of sodium hypochlorite or 70% ethanol reduce the potential risks of transmitting the virus. When in contact with patients, gloves, masks, and glasses should be used to protect against infection. According to infection control practices [112], it is necessary to follow the contact and droplet precautions, environmental cleaning and prompt response, and report clusters of cases. Of course, there is no doubt that telemedicine is effective in reducing the risk of the spread of the virus, and unnecessary eye examinations in diagnosis and management have to be avoided.

## 7. Therapy for Conjunctivitis Associated with SARS-CoV-2

There is currently no specific treatment for viral conjunctivitis, although bacterial conjunctivitis can be treated with antibacterial medication for two to three days, and the symptoms will improve. COVID conjunctivitis, like any other viral conjunctivitis, is generally self-limiting and can be managed with lubricants and symptomatic treatment, unless the cornea is involved. However, severe keratoconjunctivitis also has been reported in patients with COVID conjunctivitis [16,17]. Thus, the ocular treatment might need to be investigated.

TMPRSS2, CTSL, CD147, ADAM-17, and DPP4, in addition to ACE2, are involved in the viral entry of SARS-CoV-2. The understanding of viral entry could lead to the development of new therapeutic approaches. Most COVID-19 candidate vaccines express the spike protein or parts of the spike protein, i.e., the receptor-binding domain, as the immunogenic determinant. The vaccines that have been developed by BioNTech/Pfizer and Moderna/NIAID encode the SARS-CoV-2 spike (S) glycoprotein, stabilized in its prefusion conformation [113]. The potential treatment for COVID-19 are summarized in Table 1.

There are many types of ACE2 modulators, including recombinant soluble ACE2 and indirect ACE2 modulators such as angiotensin receptor blockers (ARBs), calmodulin antagonists, and selective estrogen receptor modifiers. Although it is not available for ocular treatment, eye drops can be less difficult for ocular application as there is some evidence that eye drops can be repurposed from systemic treatment [114]. Recombinant soluble ACE2 has already been clinically tested in acute respiratory distress syndrome [115], with phase 1 and 2 clinical trials, and, more recently, APN01 as a recombinant soluble ACE has been conducted for SARS-CoV-2 in a phase 2 clinical trial, and the study was completed as of December in 2020 (NCT04335136). As for ARBs, so far, there are more than 10 ongoing clinical trials [116]. Calmodulin (CALM) antagonists inhibit the CALM–ACE2 interaction and increase the release of the ACE2 ectodomain in a dose- and time-dependent manner [117]. Melatonin, which is known to inhibit calmodulin interaction with its target enzymes, and toremifene, which is a nonsteroidal antiestrogen that blocks the effects of estrogen, have been tested for evaluation of efficacy in several clinical trials (NCT04531748). The randomized elimination and prolongation of ACE inhibitors and ARBs in COVID-19 Trial Protocol is already underway with worldwide collaboration [118]. Similarly, use of an enhancer of ADAM-17, a metalloproteinase involved in the shedding of ACE2, can potentially work as a drug for the treatment of COVID-19 [119].

Another strategy could be the blocking of transmembrane protease TMPRSS2. Hoffmann et al. have reported that Camostat mesylate blocked the SARS-CoV-2 infection into lung cells [65]. Nafamostat, which is a drug similar to Camostat and has been widely used in clinical practice in Japan, blocked the entry process of the virus at a concentration of less than one-tenth [120]. In parallel, high throughput drug screening found that amantadine hydrochloride, which is approved by the US Food and Drug Administration for influenza and Parkinson’s disease, downregulated CTSL mRNA expression, a protease involved in spike protein activation [121]. Teicoplanin is an antibacterial drug that is already in clinical use, and it may also block SARS-CoV-2 entry through the inhibition of CTSL activity [122,123]. Targeting TMPRSS2 and Cathepsin B/L together may be synergistic against SARS-CoV-2 infection [124].

Apart from the proteases, blocking of CD147, also known as basigin (BSG) or EMMPRIN, a transmembrane glycoprotein, by the monoclonal antibody meplazumab has been found to inhibit binding of the SARS-CoV-2 spike protein [72]. Azithromycin, which is an antibiotic, induces anti-viral responses in host cells by the activation of an innate immune response via an increase in levels of interferons and interferon-stimulated proteins, and can be effective in the inhibition of SARS-CoV-2 invasion of host cells, possibly interfering with CD147/BSG [125]. A recent report claimed that no supporting evidence for a direct interaction of the SARS-CoV-2 spike protein with CD147/BSG was found [126].

The primary receptor for MERS and a predicted co-receptor for SARS-CoV-2, DPP4, a serine exopeptidase, can also be targeted for blocking the entry of the virus [79]. It has been reported that soluble DPP4 in the blood is decreased in patients with severe SARS-CoV-2 infection. A study in Italy, where the hospitalized COVID-19 patients were treated with or without the DDP4 inhibitor sitagliptin, revealed that the DPP4 inhibitor treatment improved the mortality and clinical scores, resulting in earlier hospital discharges [127].

In addition, the present topical ophthalmic medications may have potent antiviral effect. Napoli et al. performed a systematic analysis regarding the antiviral effect of topical ophthalmic medications [89]. They found that many ophthalmic eye drops have antiviral effects, including preservatives and disinfectant agents such as benzalkonium chloride and chlorhexidine, artificial tear drops, anti-glaucoma eye drops, and anti-allergy eye drops. Anti-inflammatory drugs generally have an immunosuppressive effect, but interestingly, non-steroidal anti-inflammatory drugs (NSAIDs) also have an antiviral effect. Thus, existing topical ophthalmic medications may also be effective in treating COVID conjunctivitis as drug repurposing, which is the strategy of identifying new uses for a drug that has already been approved, outside of its original medical indication. This strategy offers a number of advantages over developing an entirely new drug for one indication over an existing ophthalmic drug, and detailed future studies are needed.

## 8. Discussion

In this review, we examined the literature for factors that lead to the binding of SARS-CoV-2 on the ocular surface and we were able to show numerous factors that facilitate the entry of the virus. However, COVID conjunctivitis seems to be less common than adenovirus-related conjunctivitis, even if SARS-CoV-2 has been proven to be highly contagious. Unlike with SARS-CoV-2, the entry of adenoviruses into cells is very efficient at the ocular surface, with about 40% of the virus bound to the cell surface being able to transport its DNA to the nucleus [128]. For this reason, adenoviral conjunctivitis occurs with high frequency. Additionally, the adenoviral death protein (ADP) is synthesized very late in the infection, causing cell lysis and release of the virus from infected cells, which leads to increased infectivity [129]. But the evidence of adenovirus-related conjunctivitis and the rising cases of conjunctivitis post SARS-CoV-2 infection make a compelling case for the ocular surface as a potential site for viral entry.

Our previous systematic review revealed that one case with SARS-CoV-2 infection demonstrated keratoconjunctivitis. These symptoms impressed as pseudodendritic epithelial infiltrates, and the visual acuity decreased to 20/50 [16]. The final prognosis has not been described, but multiple subepithelial corneal infiltrates as shown in EKC could occur when the adaptive immune response starts too late, resulting in a high viral burden on the ocular surface, as Sette and Crotty proposed [130]. Recently, for the treatment of complications from multiple subepithelial corneal infiltrates following EKC, topical iodine plus steroid therapy has been proposed as an alternative treatment [131]. Drugs targeted for the replication process, except for the entry factors, would be developed in the future.

Most SARS-CoV-2 infected individuals either have mild symptoms, and/or are asymptomatic [132]. The reason for this is that any viruses, such as the family of *Adenoviridae*, activate an intracellular innate immune response associated with type I and III interferons (IFNs) after a viral infection, followed by an adaptive immune response that requires 6–10 days for proliferation and differentiation of naïve cells into effector T cells and B cells [130]. The differentiated effector immune cells protect the body by specifically eliminating infected cells and circulating viruses. On the other hand, in SARS-CoV-2 infection, the virus is known to evade or delay the activation of intracellular innate immune responses [133]. Without these responses, the virus continues to replicate and, if initiation of the adaptive immune response is too late, the innate immune system attempts to control the virus in the absence of a substantial adaptive immune response and in the presence of a high viral load, leading to an excessive inflammatory response [134,135].

In terms of replication on the ocular surface, adenoviruses and the herpes simplex virus 1 (HSV-1) are well known to replicate efficiently on the ocular surface [136,137]. In virus-infected cells, double-stranded RNA, which is produced in the cytoplasm by viral replication, acts as an inducer to induce the expression of type I and type III IFNs and IFN-stimulated genes (ISGs), resulting in intracellular antiviral activity [136,138,139,140]. Blockage of the IFN gamma (known as type III interferon) receptor (IFNgR1) increased susceptibility of the HSV-1 virus; however, interestingly SARS-CoV-2 did not replicate on the cornea [137], suggesting that SARS-CoV-2 infection is mediated by pathways other than IFNgR1 signaling, leading to delay the innate immune response.

In our review, we demonstrated that viral RNA is detectable in various tissues of the eye. Mediated by the eye’s unique environment and complicated by the reduced innate immune response to SARS-CoV-2, viral RNA can be readily transported and result in only a mild symptomatic reaction. The role of the tear film is discussed in the literature as a dynamic process that may be an essential contributor to dissemination in the context of SARS-CoV-2 [141]. The absence of symptoms with a simultaneous replication results in a problem that concerns not only ophthalmologists. Transmission can easily occur, and immunocompromised patients should wear appropriate eye protection to minimize the risk of infection. Additionally, Napoli et al. suggested the usage of eye drops and the antiviral activity of this treatment [89]. In fact, this may provide a great benefit for immunosuppressed patients and further studies should investigate this effect in more detail.

In addition, the tear turnover may play a significant role explaining the heterogeneity of the results of diagnostic studies. Understanding the possibility of SARS-CoV-2 transmission via the donor’s eyes would help to set guidelines by which to know which donor cornea can be available for corneal transplantation. In fact, procured and distributed donors have decreased since the pandemic began [101], and could lead to the potential increase of blindness due to the shortage of donor corneas.

## 9. Conclusions and Future Directions

Multiple genetic variants of SARS-CoV-2 are circulating globally. Simultaneously, multiple vaccines with high efficacy have come out in the world within just one year, which is the first such achievement ever in human history, and we have opened a new chapter in fighting infectious diseases.

With this review we can show that an understanding of the viral entry mechanisms of SARS-CoV-2 on the ocular surface can improve morbidity and mortality. Consequently, precautions should be taken to avoid the transmission of viral infectious particles from the ocular surface. Here, eye protection or also the application of eye drops plays a unique role and should be offered to clinicians and patients at risk.

A further significant aspect is the proof of the infectivity of donor corneas by an improved diagnostic. The issue of a shortage of donor corneas must be overcome soon. The mutated variants of SARS-CoV-2 could change infectivity, replication, and immune response; however, there is no need to change clinical standard precautions, and it would be essential to have a better understanding of how viruses infect and replicate in host cells and transmit on the ocular surface.

## Figures and Tables

**Figure 1 cells-10-00796-f001:**
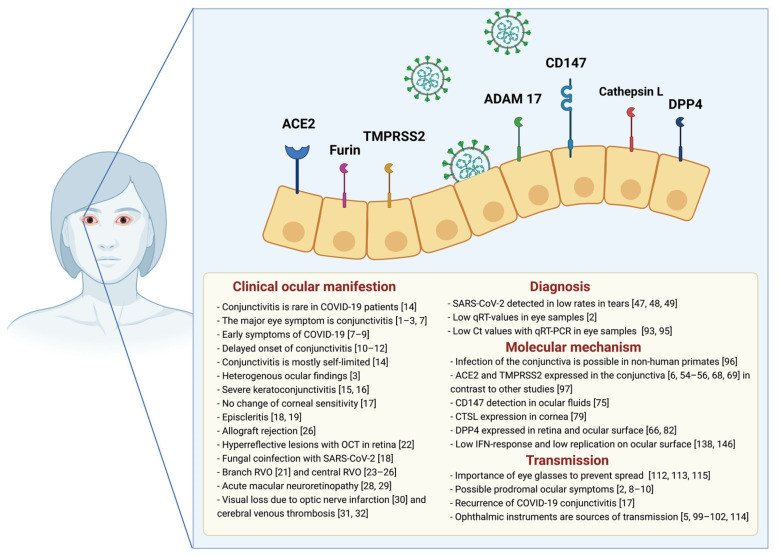
Overview of SARS-CoV-2 entry molecules, clinical manifestations, and transmission on the ocular surface with references. Figure created with BioRender.com. COVID-19: Coronavirus disease 2019, SARS-CoV-2: Severe acute respiratory syndrome coronavirus 2, OCT: Optical Coherence Tomography, RVO: Retinal Vein Occlusion, qRT-PCR: Quantitative reverse transcription polymerase chain reaction, Ct: Cycle threshold, ACE2: Angiotensin-converting enzyme 2, TMPRSS2: Transmembrane serine protease 2, CTSL: Cathepsin L, DPP4: Dipeptidyl peptidase-4, IFN: Interferon.

**Table 1 cells-10-00796-t001:** Potential drugs for the treatment of SARS-CoV-2.

Name of Potential Drugs	Targets	Function
APN01	ACE2	Soluble ACE2 [115] [NCT04335136]
Angiotensin receptor blocker	ACE2	ACE2 modulator [116]
Melatonin	ACE2	Calmodulin inhibitor [117]
Toremifene	ACE2	Nonsteroidal antiestrogen [NCT04531748]
5-fluorouracil	ADAM-17	ADAM-17 enhancer [119]
Camostat mesylate	TMPRSS	TMPRSS inhibitor [65]
Nafamostat mesylate	TMPRSS	TMPRSS inhibitor [120]
Amantadine	CTSL	CTSL inhibitor [121]
Teicoplanin	CTSL	CTSL inhibitor [122,123]
Meplazumab	CD147	Anti-CD147 [NCT04275245]
Azithromycin	CD147	Indirect CD147 inhibition [125]
Sitagliptin	DPP4	DDP4 inhibitor [127]

## Data Availability

Not applicable.

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
