# Peer review of "The Transmission of SARS-CoV-2 Infection on the Ocular Surface and Prevention Strategies"

_cells, 2021, doi:10.3390/cells10040796_

Round 1

Reviewer 1 Report

I read with interest the article entitled “The clinical impact of viral entry of SARS-CoV-2 on the ocular surface” attempting to investigate the interaction between SARS-CoV-2 and  the eye.

However, several flaws and questions are raised in this manuscript that need to be addressed and clarified. In particular, the discussion should be made more robust and convincing by means of a more thorough bibliographic research. Adequate and thorough literature search is essential for a similar article (currently, we have more than 111,000 articles about COVID-19!!). In this sense, different important papers on COVID-19 can reinforce the present article. Moreover, the text is very similar to other articles already published.

  1. The article reports: “there have still been limited data with reports to the eye, and it is controversially debated if SARS-CoV-2 can infect the ocular surface and transmit via the ocular fluid.” Please note that this issue was widely discussed in the article entitled “The Ocular Surface and the Coronavirus Disease 2019: Does a Dual ‘Ocular Route’ Exist?” J. Clin. Med. 2020, 9, 1269.” Clearly, all the potential ocular routes of SARS-CoV-2 transmission are important issues regarding the pandemic disease, as well as the related personal protective equipment (e.g. eye goggles). Eye protection should be recommended for health care workers and patients in particular situations. Please cite the aforementioned article (in compliance with the rules n. 6-7 of the COPE guidelines) and discuss this point. This reinforces the article.
  2. The authors also mention the problems regarding the susceptibility to SARS-CoV-2 infection for ophthalmologists: “The instruments for eye examinations, including slit lamps and their accessory lenses, have also been known to be a source of viral transmission”. This important topic should be explored a little. Even in this case, the authors should implement literature search, and mention the lack of universally agreed recommendations on safety systems (and legal protection) for clinical ophthalmologists and eye surgeons: “Safety Recommendations and Medical Liability in Ocular Surgery during the COVID-19 Pandemic: An Unsolved Dilemma.” J. Clin. Med. 2020, 9, 1403. (Napoli, P.E.; et al.). In this way, the authors may highlight this unresolved health care problem.
  3. The authors mention various studies reporting controversial data concerning the different percentages of positive results for SARS-CoV-2 in tear samples from COVID-19 patients. The authors should note that the majority of these studies did not report/investigate the use of eye drops in these patients. However, a large number of ophthalmic medications has antiviral action: “A Panel of Broad-Spectrum Antivirals in Topical Ophthalmic Medications from the Drug Repurposing Approach during and after the Coronavirus Disease 2019 Era. J. Clin. Med. 2020, 9, 2441.“ Please quote and discuss this potential bias.
  4. The title reports and seems announce the theme of clinical impact of SARS-CoV-2 on the ocular surface. However, the aims of this review appear to be almost vague and heterogeneous. Probably the authors should reconsider the title.
  5. The methods are explained rather superficially. Authors should better explain the selection (inclusion and exclusion) of the articles.
  1. Please clarify the aims of the article in the abstract.

  1. The article report “We highlight findings that SARS-CoV-2 RNA and its spike protein were detected in the various ocular tissues from patients and donor eyes with COVID-19, however, the expression level seemed to be relatively low compared to the respiratory tissues. This may be due to the unique environment surrounding the ocular surface and the innate immune response of SARS-CoV-2, which could affect the viral replication following the viral entry into the ocular cells.“ Please note that the dynamics of the tear film could carry the virus towards the mucous membrane of the nose and onto the skin of the face. Please improve the discussion of this point based on the following article: “Fourier-Domain OCT Imaging of the Ocular Surface and Tear Film Dynamics: A Review of the State of the Art and an Integrative Model of the Tear Behavior during the Inter-Blink Period and Visual Fixation. Clin. Med. 2020, 9, 668.”

  1. Important concern: What is the usefulness of this article?What are the original findings of this research article compared to the others?Please clarify.

In sum, the article is quite interesting, but discussion should be improved as suggested. At the moment, the paper is not ready for publication. I have some concerns about some areas noted above, particularly in bibliographic research, and I would like to see the revised version again before allowing it to be published.

Author Response

Comment 1: I read with interest the article entitled “The clinical impact of viral entry of SARS-CoV-2 on the ocular surface” attempting to investigate the interaction between SARS-CoV-2 and the eye.

However, several flaws and questions are raised in this manuscript that needs to be addressed and clarified. In particular, the discussion should be made more robust and convincing by means of more thorough bibliographic research. An adequate and thorough literature search is essential for a similar article (currently, we have more than 111,000 articles about COVID-19!!). In this sense, different important papers on COVID-19 can reinforce the present article. Moreover, the text is very similar to other articles already published.

Response 1: We greatly appreciate the Reviewer’s insightful comment. As the reviewer pointed out, we have realized that the aim of our review was not yet clear and the discussion was unclear, making it difficult for the reader to understand this review. The purpose of this review article has been clearly stated in the introduction, the method has been written correctly, and an extensive literature search has been conducted and reviewed again according to the inclusion and the exclusion criteria. Please find below our point-by-point responses to your thoughtful suggestions.

Comment 2: The article reports: “there have still been limited data with reports to the eye, and it is controversially debated if SARS-CoV-2 can infect the ocular surface and transmit via the ocular fluid.” Please note that this issue was widely discussed in the article entitled “The Ocular Surface and the Coronavirus Disease 2019: Does a Dual ‘Ocular Route’ Exist?” J. Clin. Med. 2020, 9, 1269.” Clearly, all the potential ocular routes of SARS-CoV-2 transmission are important issues regarding the pandemic disease, as well as the related personal protective equipment (e.g. eye goggles). Eye protection should be recommended for health care workers and patients in particular situations. Please cite the aforementioned article (in compliance with the rules n. 6-7 of the COPE guidelines) and discuss this point. This reinforces the article.

Response 2: We greatly appreciate the Reviewer’s insightful comment. We also totally agree with the reviewer. As the reviewer suggested, the majority of COVID-19 studies did not report and investigate the use of eye drops, which could have a potent antiviral effect, that has been reported by Napoli et al. Many researchers also proposed that health care workers and patients needed to protect the eyes from SARS-CoV-2, since the eye is considered to be a potential route of SARS-CoV-2 transmission.

At the same time, however, we would say that the eyes are not a common route of SARS-CoV2. We now believe that the eyes are, of course, an important potential route of transmission of SARS-CoV-2, but they are not a common route, and there are many controversial data as well. However, this does not mean that we should not protect our eyes. In compliance with the COPE guidelines, we have added more citations and explanations to make it clear. (Revised Manuscript, lines 62-64, 71-72, 306-311, 403-405, 562-565)

Comment 3: The authors also mention the problems regarding the susceptibility to SARS-CoV-2 infection for ophthalmologists: “The instruments for eye examinations, including slit lamps and their accessory lenses, have also been known to be a source of viral transmission”. This important topic should be explored a little. Even in this case, the authors should implement literature search, and mention the lack of universally agreed recommendations on safety systems (and legal protection) for clinical ophthalmologists and eye surgeons: “Safety Recommendations and Medical Liability in Ocular Surgery during the COVID-19 Pandemic: An Unsolved Dilemma.” J. Clin. Med. 2020, 9, 1403. (Napoli, P.E.; et al.). In this way, the authors may highlight this unresolved health care problem.

Response 3: We greatly appreciate the Reviewer’s insightful comment. We also believe that it is necessary to address the issue regarding the susceptibility to SARS-CoV-2 infection to ophthalmologists. As the Reviewer suggested, we have added more literature through extensive literature search correctly including a paper reported by Napoli, et al. in a journal of J Clin Med. (Revised Manuscript, lines 350-355, 357-364, 369-372, 375-395)

Comment 4: The authors mention various studies reporting controversial data concerning the different percentages of positive results for SARS-CoV-2 in tear samples from COVID-19 patients. The authors should note that the majority of these studies did not report/investigate the use of eye drops in these patients. However, a large number of ophthalmic medications has antiviral action: “A Panel of Broad-Spectrum Antivirals in Topical Ophthalmic Medications from the Drug Repurposing Approach during and after the Coronavirus Disease 2019 Era. J. Clin. Med. 2020, 9, 2441.“ Please quote and discuss this potential bias.

Response 4: We want to thank the reviewer for this helpful comment. As we have mentioned in Response 2, we have now realized that the majority of COVID-19 studies did not report and investigate the use of eye drops, which could have a potent antiviral effect, that has been reported by Napoli et al. We have cited a paper published in a journal of J Clin Med. 2020, 9, 2441. and have described the bias of the study that did not investigate the use of eye drops. (Revised Manuscript, line 481-492)

Comment 5: The title reports and seems announce the theme of clinical impact of SARS-CoV-2 on the ocular surface. However, the aims of this review appear to be almost vague and heterogeneous. Probably the authors should reconsider the title.

Response 5: We greatly appreciate the Reviewer’s helpful suggestion. We have now realized that the aim of our review was not yet clear and the discussion was unclear, making it difficult for the reader to understand. In order to make the purpose of our review article clear, we have revised the title to “The transmission of SARS-CoV-2 infection on the ocular surface and prevention strategies”. (Revised Manuscript, lines 1-2)

Comment 6: The methods are explained rather superficially. Authors should better explain the selection (inclusion and exclusion) of the articles.

Response 6: We greatly appreciate the Reviewer’s helpful comment. We have conducted an extensive literature search according to precisely inclusion and exclusion criteria, and have described the method for our literature search in the Method section. (Revised Manuscript, lines 164-176)

Comment 7: Please clarify the aims of the article in the abstract.

Response 7: We greatly appreciate the Reviewer’s helpful comment. We have now revised the title of this article and abstract to make the aims of our review article clear. (Revised Manuscript, lines 18-30)

Comment 8: The article report “We highlight findings that SARS-CoV-2 RNA and its spike protein were detected in the various ocular tissues from patients and donor eyes with COVID-19, however, the expression level seemed to be relatively low compared to the respiratory tissues. This may be due to the unique environment surrounding the ocular surface and the innate immune response of SARS-CoV-2, which could affect the viral replication following the viral entry into the ocular cells.“ Please note that the dynamics of the tear film could carry the virus towards the mucous membrane of the nose and onto the skin of the face. Please improve the discussion of this point based on the following article: “Fourier-Domain OCT Imaging of the Ocular Surface and Tear Film Dynamics: A Review of the State of the Art and an Integrative Model of the Tear Behavior during the Inter-Blink Period and Visual Fixation. Clin. Med. 2020, 9, 668.”

Response 8: We greatly appreciate the Reviewer’s insightful comment. As the reviewer points out, it can be inferred that tear fluid dynamics, in addition to the innate immune response on the ocular surface, influence the entry of SARS-CoV-2 to the ocular surface cells. In fact, the virus entering through the eyes would be immediately carried out by the tear fluid to the mucous membrane of the nose and onto the skin of the face. We have discussed on this point with referring to the following paper; Clin. Med. 2020, 9, 668. (Revised Manuscript, lines 541-543, 549-550).

Comment 9: Important concern: What is the usefulness of this article? What are the original findings of this research article compared to the others? Please clarify.

Response 9: We greatly appreciate the Reviewer’s insightful comment. We have now realized that the aims of our review were vague and unclear. So, we have reorganized the whole structure to make readers understand more easily. (Revised Manuscript, lines 496-498, 538-554).

Comment 10: In sum, the article is quite interesting, but discussion should be improved as suggested. At the moment, the paper is not ready for publication. I have some concerns about some areas noted above, particularly in bibliographic research, and I would like to see the revised version again before allowing it to be published.

Response 10: We greatly appreciate the Reviewer’s cogently helpful comment. We have added the inclusion and exclusion criteria regarding extensive literature search in Method section, and reorganized the entire structure of our review article to more focus on the eye involvement of SARS-CoV-2.

 (Revised Manuscript, lines 56-68, 164-176).

Reviewer 2 Report

General Comments

Lines 47-50:

“Approximately 70% of patients with acute conjunctivitis present to primary and urgent care 48 [5], and conjunctivitis is a common presenting problem for general practitioners (GPs) 49 despite the likely low risk of viral shedding from the conjunctiva.”

Is this SARS-CoV2 conjunctivitis or any type of viral and/or bacterial conjunctivitis?

Line 133:

Language is a little confusing - change ‘can prevent’ to ‘are resistant to’

Lines 151 – 152

Language is a little confusing – perhaps change ‘Independent authors have published an increased viral load triggered by the D614G mutation [40,41]’ to ‘Independent authors have determined that there is an increased viral load triggered by the D614G mutation’

Lines 365 – 366:

The sentence ‘The viral entry of SARS-CoV-2 is involved in TMPRSS2, CTSL, CD147, ADAM-17 and DPP4 in addition to ACE2’ seems to be backwards. Perhaps it’s best to say ‘TMPRSS2, CTSL, CD147, ADAM-17 and DPP4, in addition to ACE2, is involved in the viral entry of SARS-CoV-2’.

Author Response

Comment 1: Lines 47-50: “Approximately 70% of patients with acute conjunctivitis present to primary and urgent care 48 [5], and conjunctivitis is a common presenting problem for general practitioners (GPs) 49 despite the likely low risk of viral shedding from the conjunctiva.”

Is this SARS-CoV2 conjunctivitis or any type of viral and/or bacterial conjunctivitis?

Response 1: We greatly appreciate the Reviewer’s cogently helpful comment. We would like to apologize for the inconclusive section. This was mentioned about general viral conjunctivitis. But we have now realized that this description has not been suitable in this COCID-19 paper. So, we have omitted it in Introduction section.

Comment 2: Line 133: Language is a little confusing - change ‘can prevent’ to ‘are resistant to’

Response 2: We greatly appreciate the Reviewer’s cogently helpful comment. We have revised the reviewer’s pointed word. (Revised Manuscript, line 140).

Comment 3: Lines 151 – 152: Language is a little confusing – perhaps change ‘Independent authors have published an increased viral load triggered by the D614G mutation [40,41]’ to ‘Independent authors have determined that there is an increased viral load triggered by the D614G mutation’

Response 3: We greatly appreciate the Reviewer’s cogently helpful comment. We have revised the reviewer’s pointed sentence. (Revised Manuscript, lines 158-159).

Comment 4: Lines 365 – 366: The sentence ‘The viral entry of SARS-CoV-2 is involved in TMPRSS2, CTSL, CD147, ADAM-17 and DPP4 in addition to ACE2’ seems to be backwards. Perhaps it’s best to say ‘TMPRSS2, CTSL, CD147, ADAM-17 and DPP4, in addition to ACE2, is involved in the viral entry of SARS-CoV-2’.

Response 4: We greatly appreciate the Reviewer’s cogently helpful comment. I have revised the reviewer’s pointed sentence. (Revised Manuscript, lines 430-431).

Reviewer 3 Report

The Authors report an extensive review on the molecular mechanisms of action of SARS COV 2, with particular focus on the ocular surface involvement. The Manuscript is interesting, however sometimes it loses the focus on the eye. Some paragraphs need to be more synthetic and focused. I suggest to review the Title since the focus is not on the clinical impact but on the mechanisms of ocular entry and involvement. I also suggest to review the entire article because there is a need of a more focused article on the eye.  

Author Response

Comment 1: The Authors report an extensive review on the molecular mechanisms of action of SARS COV 2, with particular focus on the ocular surface involvement. The Manuscript is interesting, however sometimes it loses the focus on the eye. Some paragraphs need to be more synthetic and focused. I suggest to review the Title since the focus is not on the clinical impact but on the mechanisms of ocular entry and involvement. I also suggest to review the entire article because there is a need of a more focused article on the eye.

Response 1: We greatly appreciate the Reviewer’s insightful comment. We have reviewed and reorganized the entire structure of our review article.

We have described the aims of this review in the Introduction section to more focus on the eye involvement of SARS-CoV-2 to make it clear, and have added the inclusion and exclusion criteria regarding extensive literature search in Method section.

The aim of our review was to clarify the importance of SARS-CoV-2 infection of the eye for transmission and protection through the understanding of the entry factor of SARS-CoV-2. To clarify this, we have made changes to the Abstract, Introduction and Discussion, and Outlook.

Our main findings in this review are:

1) a considerable number of receptors are involved in the viral entry of SARS-CoV-2 on the eye (TMPRSS2, CTSL, CD147, ADAM-17 and DPP4, in addition to ACE2)

2) the ocular surface is a potential route of transmission for SARS-CoV-2, although not common.

3) However, it is clear that a transmission even without symptoms can occur. Based on this, an eye protection is strongly recommended for ophthalmologists and patients with risk factors. Furthermore, the eye drop approach is very well suited and should definitely be investigated in further studies.

We have extensively discussed the findings noted above throughout this review.

Round 2

Reviewer 3 Report

The authors addressed well all reviewer's requests. 

This manuscript is a resubmission of an earlier submission. The following is a list of the peer review reports and author responses from that submission.